# Aqueous Extract of *Phyllanthus emblica* L. Alleviates Functional Dyspepsia through Regulating Gastrointestinal Hormones and Gut Microbiome In Vivo

**DOI:** 10.3390/foods11101491

**Published:** 2022-05-20

**Authors:** Xiaoqing Li, Yilin Lin, Yiqi Jiang, Binbin Wu, Yigang Yu

**Affiliations:** 1College of Food Science and Engineering, South China University of Technology, Guangzhou 510006, China; lixiaoqing318@163.com (X.L.); echoyilin@126.com (Y.L.); jiangyiqizjk@163.com (Y.J.); 2Lui Che Woo Institute of Innovative Medicine, Hong Kong Hub of Paediatric Excellence (HK HOPE), The Chinese University of Hong Kong (CUHK), Hong Kong SAR, China; binbinwu@cuhk.edu.hk

**Keywords:** aqueous *Phyllanthus emblica* L. extract, functional dyspepsia, gastrointestinal hormones, gut microbiome

## Abstract

*Phyllanthus emblica* L. fruits were extracted by a hot water assistant with ultrasonication to obtain aqueous *Phyllanthus emblica* L. extract (APE). The ameliorating functional dyspepsia (FD) effect of a low dose (150 mg/kg) and a high dose (300 mg/kg) of APE was exhibited by determining the gastrointestinal motility, gastrointestinal hormones, and gut microbiome shifts in reserpine induced FD male balb/c mice. APE increased the gastrointestinal motility including the gastric emptying (GE) rate and small intestinal transit (SIT) rate. The level of serum gastrointestinal hormones such as motilin (MTL) and gastrin (GAS) increased, and the vasoactive intestinal peptide (VIP) level decreased after the administration of APE. Furthermore, the gut microbiome analysis demonstrated that APE could regulate the microbiome structure and restore homeostasis by elevating useful bacterial abundance, while simultaneously decreasing harmful bacterial abundance. This study demonstrated the ameliorating FD effect of APE and its potential efficacy in curing functional gastrointestinal disorders and maintaining a healthy digestive tract.

## 1. Introduction

Defined by epigastric symptoms, functional dyspepsia (FD) is a chronic or intermittent gastrointestinal disorder and gastric sensation, or gastric and duodenal inflammation, which is believed to originate from the gastroduodenal region [1]. According to the Rome IV Diagnostic Criteria published in 2016, FD can be characterized by four main symptoms: postprandial fullness, early satiation, upper abdominal pain, and upper abdominal burning. Furthermore, extra-intestinal symptoms are prevalent in FD, including fatigue, insomnia, and pain outside the gastrointestinal tract, which could not be explained until recently [2]. Although FD may not threaten one’s life, one’s quality of life can be seriously influenced in nearly all patients. There is a significant difference in the prevalence and subtypes in different countries due to gender, race, diet, and diagnostic criteria; the prevalence has been reported to range from 5% to 20% [3,4]. FD has been regarded as a multifactorial functional gastrointestinal dysfunction in which gastric motor disorder, mental factors, gastroduodenal sensitivity, mucosal inflammation, intestinal flora, and gastrointestinal hormones have all been considered to play a part [1,5,6]. Currently, plenty of therapeutic methods are applied for the treatment of FD, including the eradication of *Helicobacter pylori*, antidepressant therapy, prokinetic therapy, acid inhibition therapy, probiotic therapy, and traditional herbal medicinal therapy [7,8,9]. In consideration of the adverse effects and unsatisfactory results [10], searching for healthy therapies and natural substances with alleviating FD effects has recently attracted a great deal of attention. Some herbal and natural materials have been verified to relieve the symptoms of FD and exert a wider range of pharmacological effects when compared with targeted therapy [9,11,12,13].

*Phyllanthus emblica* L., known as Amla or Indian Gooseberry, belongs to the family Euphorbiaceae. It is a traditional medical plant that has been widely used to treat diseases in some tropical and subtropical countries such as India, China, Pakistan, Uzbekistan, Sri Lanka, and Malaysia since antiquity [14]. Amla is one of three healthcare plants recommended by the WHO and one that is also recorded in Chinese pharmacopoeia (2010) [15]. It can be used as a herbal medicine as well as a kind of delicious food.

Amla is a rich source of nutrients. There are hundreds of chemical constituents in Amla, commonly containing phenolic acids, tannins, alkaloids, terpenes, sterols, fatty acids, flavonoids, amino acids, vitamin C, and other compounds, demonstrating its diversity in biological activities [16,17]. Many researchers have proven that Amla possesses antidiabetic [18], gastroprotective [19], hepatoprotective [20,21], anti-inflammatory [15], anti-hyperlipidemic [22], antitussive [23], antimicrobial [24], antioxidant [25], anticancer [26], and antiatherogenic functions [27]. Moreover, as a digestive, Amla was reportedly capable of preventing peptic ulcers and dyspepsia [14,28]. In this research, we investigated the possible mitigating FD effect of aqueous *Phyllanthus emblica* L. extract by detecting changes in gastrointestinal motility, gastrointestinal hormones, and the gut microbiome.

## 2. Materials and Methods

### 2.1. Preparation and Identification of Aqueous Extract from Phyllanthus Emblica

The fresh *Phyllanthus emblica* fruits were purchased from Yunnan Tianqi Biotechnology Co., LTD (Baoshan, China), which were collected from Yunnan province in southwest China in July and heated until dry by an air-dry oven at 55 °C. The 20 g dry fruits without stones were smashed to a powder and were extracted using distilled water twice at 65 °C with an ultrasonication frequency of 55 kHz. The solid–liquid ratio was 1:40. The supernatant was collected and evaporated after the extract had been filtered and centrifuged at 5000 rpm for 15 min. The resultant liquid was then lyophilized to obtain 11 g of the final aqueous *Phyllanthus emblica* extract (APE). The total phenolic content of APE was determined using a modified Folin–Ciocalteu method and gallic acid was used as a standard [29]. The total sugar content was measured by a typical phonel-sulfate method. The HLPC fingerprint of APE was conducted by a SHIMADZU 15C HPLC-PDA system equipped with a DiKMA C18 column (250 mm × 4.6 mm × 5 μm). The mobile phase was 0.1% aqueous methanoic acid (A) and methanol (B) at a flow rate of 1 mL/min with the program as follows: 0–6 min, 5–7% B; 6–16 min, 7–30% B; and 16–51 min, 30–80% B. The chromatogram was monitored at a wavelength range from 200 nm to 800 nm. The injection volume was 10 μL.

### 2.2. Animal Treatment

Forty male balb/c mice (18–22 g, 6 weeks old) were delivered from Guangdong SijiaJingda Biological Technology Co., LTD (Guangzhou, China) with the certification number SCXK2020-0052. The mice were maintained in a stable Specific Pathogen Free room with a consistent room temperature (20–24 °C) at the Laboratory Animal Center of South China Agricultural University. All animal procedures were carried out in accordance with the ethical review board of South China Agriculture University, with the certification number SCAU-2021B169, and was approved on 30 September 2021.

This experiment adopted the previously validated mice model of FD using the method of injecting reserpine (1 mg/kg/d) intraperitoneally combined with fasting every other day for two weeks [9,11,12,30]. All mice were accommodated for seven days and then divided into four groups including a normal group (N), a model group (M), an APEL group (150 mg/kg), and an APEH group (300 mg/kg). All groups were induced into the FD model with the reserpine injection and an irregular diet, except for the normal group which was fed a normal diet every day. After two weeks of modeling, the APEL group and the APEH group were orally administered a low and high dose of APE, while the normal and model groups were treated with the same volume of saline. Food intake and body weight were weighed and recorded every day. In the process, body shape, appetite, activity, hair, and defecation were observed. After three weeks of treatment, all mice were fasted for 24 h and then sacrificed. Blood, stomach, and intestinal content were collected and stored for the follow-up experiment.

### 2.3. Measurement and Analysis of Gastric Emptying (GE) Rate and Small Intestinal Transit Rate (SIT)

GE and SIT experiments were carried out according to the previous method with a small modification [31]. In brief, every mouse was given 0.6 mL of semi-solid paste with carbon powder by intragastric gavage, which consisted of 5 g of sodium carboxymethyl cellulose, 4 g of starch, 8 g of milk powder, 4 g of sugar, and 4 g of activated carbon with a total volume of 150 mL after overnight fasting. All mice were sacrificed after 30 min, and the stomach and small intestine were quickly isolated. The stomach was weighed the first time to record the total stomach mass. Afterwards, 0.9% saline was used to remove the residue inside of the stomach and then weighed a second time and recorded as an empty stomach mass. The whole section of the small intestine from the upper pylorus and the lower ileocecal part was taken and carefully removed, and then laid on white paper without traction. The distance from the pylorus to the front of the black semi-solid paste and the total length of the small intestine were measured with a ruler. The formulas to calculate GE ratio and SIT ratio were as follows:GE ratio (%) = [1−(total stomach mass − empty stomach mass)/total stomach mass]×100%
SIT ratio (%)= Small intestine propulsion length/total length of small intestine ×100%

### 2.4. Detection of Serum Gastrointestinal Hormones

The whole blood was stored at 4 °C for 2 h and then centrifuged at 4 °C (3500× *g*, 7 min) to obtain the serum. Gastrointestinal hormones including motilin (MTL), gastrin (GAS), cholecystokinin (CCK), and vasoactive intestinal peptide (VIP) were determined via Enzyme-Linked Immunosorbent Assay (ELISA) kits (Jiancheng Technology Co., Ltd., Nanjing, China) according to the manufacturer’s instructions.

### 2.5. Microbiome Analysis

The Bacterial DNA of the intestinal content was extracted with an Invitrogen Pure Link Microbiome DNA Purification Kit (Thermo Fisher, Waltham, MA, USA). The extracted DNA was quantified by Nanodrop (Thermo, Waltham, MA, USA), and then detected by a 1.2% agarose gel electrophoresis. PCR amplicon was performed in V3+V4 fragments of 16S rRNA, using universal primers as follows: 338F (ACTCCTACGGGAGGCAG CAG) and 806R (GGACTACHVGGGTWTCTAAT). The amplified DNA products were purified by Vazyme VAHTSTM DNA clean beads and quantified by a Microplate reader (BioTek, FLx800) with a fluorescent reagent of Quant-iT PicoGreen dsDNA Assay Kit. Bacterial DNA was monitored by an Illumina MiSeq PE250 high-throughput sequencing platform with a MiSeq Reagent Kit V3 (600 cycles). Sequence processing or operational taxonomic unit (OTU) clustering was performed according to a QIIME2 DADA2 analysis. The relative abundance of the bacteria and linear discriminant analysis effect size (LEfSe) were used to explore the different intestinal microbiomes. The correlation analysis was based on a Spearman index determined by R software (version 4.0); the package was used to detect the links between intestinal flora and gastrointestinal hormones.

### 2.6. Statistical Analysis

The data are presented as the mean ± standard deviation. A one-way analysis was used to compare the significant differences between groups with the GraphPad Prism statistical software (version 8.0.2) (San Diego, CA, USA). Furthermore, *p* < 0.05 was taken as statistically significant difference.

## 3. Results and Discussion

### 3.1. Chemical Composition Characterization of APE

The final yield of APE from the dry Phyllanthus emblica fruit hull was 55%. The total sugar content and phenolic content of APE were 23.6% and 30.77%, respectively. Four chemical compounds including gallic acid, corilagin, ellagic acid, and fistin were used as reference standards to characterize APE by HPLC analysis qualitatively. The existence of four compounds was confirmed in APE and the representative spectrums under 280 nm are exhibited in Appendix A.

### 3.2. Food Intake and Body Weight

After modeling for two weeks, mice in the model, APEL, and APEH groups showed signs of squinting, irritability, crouching, sensitivity to environmental changes, frequent wrestling amongst themselves, changes in stool from dry to soft, weight loss, and decreased food intake. After the administration of APE for three weeks, the body weight of APE groups increased and eventually reached the weight of the normal group, which showed that APE could help the body weight recovery of FD mice (Figure 1A). After the reserpine injection and irregular diet, the food intake dropped in the modelling group when compared to the normal group. The food intake of the APE group increased faster than that of the model group and finally reached the levels of the normal group, demonstrating that APE is beneficial for the appetite and the restoration of food intake (Figure 1B).

### 3.3. Gastrointestinal Motility

Gastrointestinal digestive motility is an important index to evaluate digestive function. FD is usually characterized by gastrointestinal motility disorder [32] and delayed GE rate [33]. Therefore, GE rate and SIT rate can be used as indicators to assess digestive function [34]. As shown in Figure 2, when compared with the normal group, GE rate and SIT rate in the model control group were significantly reduced, indicating that the modeling was successful. Compared to the model group, the gastric empty rate and small intestinal transit rate in the APEL and APEH groups were significantly higher, indicating that APE could improve the delayed GE rate and SIT rate caused by FD.

### 3.4. Serum Gastrointestinal Hormones

Gastrointestinal hormones are a group of peptides distributed in the gastrointestinal tract and blood circulation. Some are related to gastrointestinal motility, for instance, GAS, MTL, VIP, and CCK [11]. MTL was verified to be capable of contracting the smooth muscles of the gastrointestinal tract to promote gastrointestinal motility and GE rate. GAS is an acid-stimulatory messenger that can regulate gastric acid secretion and gastric mucosal cell growth, thereby increasing the food intake [35]. CCK is secreted from intestinal I cells after high-lipid food is taken in. It can inhibit GAS release and slow GE through the inhibition of antral contractility and the stimulation of pyloric contractions [36,37]. VIP inhibits gastrointestinal motility and gastric acid secretion [38]. The mice serum GAS, MTL, VIP, and CCK were evaluated after treatment. As shown in Figure 3, MTL and GAS levels in low and high doses of the APE group significantly increased compared to the model group, indicating that APE could effectively promote gastric contraction and movement, as well as upper digestive tract movement (Figure 3A,B). Moreover, compared to the model group, serum VIP content in APE groups significantly decreased, even reaching the level of the normal group. However, the APE treatment did not change CCK levels, indicating that CCK may not be involved in the efficacy of APE against delayed gastrointestinal motility induced by FD. It was supposed that APE played a part in the stimulation of gastrointestinal motility and digestion efficiency by regulating the homeostasis of gastrointestinal hormones.

### 3.5. Microbiome Analysis

Accumulating evidence shows that the homeostasis of intestinal bacteria has a strong connection with gastrointestinal function [39]. The intestinal bacteria exert multiple guardian and metabolic effects in the intestines and play a large part in maintaining gut homeostasis [40]. Previous research showed that the structure and diversity of intestinal microbiota were substantially different in gastrointestinal disorder patients when compared to healthy ones [41]. In this study, intestinal microbiota was analyzed to reveal the effect of APE on FD mice by regulating the microbiome structure and quantity. The entire bacterial DNA of the intestinal content was extracted and sequenced. After denoising and removing the unreasonable data, the length distribution of the high-quality fecal bacteria sequences contained in all samples mainly concentrated around 400 bp (Figure 4A). The OTUs were clustered and divided with a 97% sequence similarity. There were 1003 OTUs shared by four groups and 343 OTUs overlapped in the N, APEL, and APEH groups. However, there were 256 OTUs shared by the M, APEL, and APEH groups, indicating that the microbiome in APE groups was more like normal groups when compared to the model group (Figure 4B). The Chao1 and Shannon indices were used to assess the α-diversity of the gut microbiome. Two diversity indices showed that APE treatment in an especially high-dosage group improved the diversity and richness of the microbiome community (Figure 4C,D). Distinct differences in the flora constitution among the normal and model groups were observed from the principal component analysis (PCA). The administration of APE ameliorated this change and indicated that APE could relieve the reduced gut microbial diversity in FD mice (Figure 4E).

The relative richness at the phylum level indicated that Firmicutes and Bacteroidetes were the two primary phyla among the four groups (Figure 5A). An analysis of the abundance of these two main phyla showed that Firmicutes decreased in the model group in comparison to the normal group and the administration of APE reversed this trend in a dose-dependent manner. On the contrary, the abundance of Bacteroidetes was boosted in the model group and the low dose of APE treatment significantly reduced it (Figure 5C,D). It was generally accepted that gram-negative Bacteroidetes had the potential to promote inflammation, and some are known to be opportunistic pathogens. A previous study had demonstrated that Proteobacteria levels were higher in the gut mucosa of IBS patients compared to healthy ones [42]. The reduction in Firmicutes was also reported in high fat diet induced mice [43]. At the genus level (Figure 5B), the genera *Muribaculaceae*, *Alistipes,* and *[Eubacterium]_xylanophilum_group* rose in the FD mice declined after APE treatment (Figure 5E,G). The richness of the genera *Lactobacillus* (Figure 5F) and *Faecalibaculum* showed an opposite tendency that decreased in the model group and was restored in the APE groups. It was reported that *Muribaculaceae* dominated the intestinal flora of mice in the high-fat diet group and exhibited lower sensitivity to the bactericidal antibiotic [44]. The Genera *[Eubacterium]_xylanophilum_group* was verified to be positively related to branched-chain amino acid levels, which is the diagnosis index of obesity [45]. The increased pathogenic bacteria *Alistipes* with impaired gut barrier function were associated with colorectal tumorigenesis [46]. *Lactobacillus* and *Faecalibaculum* was proven to be related to the production of short-chain fatty acids [47]. In conclusion, the use of APE changed the status of the FD-induced mice, decreasing harmful bacteria and elevating beneficial bacteria, while helping to restore the microbiome balance.

The LEfSe based on linear discriminant analysis was used to evaluate the structure of gut microbiota from the phylum level to the genus level (Figure 6A,B). A large shift in the microbiome abundance was evident from the result. The genus *Prevotellaceae* was dominant in the model group, while *Bacterioides* and *Akkermansia* were dominant in the normal group. Additionally, c_Clostridia, f_Lachnospiraceae, g_Roseburia, g_Jeotgalicoccus, and c_Erysipelotrichales were more abundant in the APEL and APEH groups. The alterations of the flora population might be related to a change in the biochemical criterion. To reveal the underlying relationship in the gut flora structure and gastrointestinal indices, a correlation between the top 20 abundant gut microorganisms at the genera level and six gastrointestinal indices were analyzed based on Spearman’s rank-order. *Muribaculaceae* and *Alistipes* negatively correlated with most indexes except serum VIP, indicating that the decreased proportion of *Muribaculaceae* and *Alistipes* could help to promote gastrointestinal movement and relieve the FD symptoms. On the contrary, the ratio of *Lactobacillus* and *Faecalibaculum* positively correlated with all five gastrointestinal indices except serum VIP, indicating that the growing of these two genera had a beneficial effect on gastrointestinal disorder. All the correlation analysis results were consistent with the relative abundance variation of the microbial community at the genus level. The alteration in the relative bacterial abundance was related to the gastrointestinal motility and serum gastrointestinal hormone level. Chemical composition characterization showed that APE is rich with polyphenol substances and saccharides. Previous researchers have shown that bioactive substances such as polyphenol and polysaccharides improve diseases through modulating the composition of gut microbiota [31,48]. From the above results, the conclusion can be drawn that APE may have exerted its effect on FD through improving the relative bacterial abundance of the gut microbiome by increasing the proportion of effective microbes, such as *Lactobacillus* and *Faecalibaculum*, while decreasing harmful bacteria, such as *Muribaculaceae*, *Alistipes*, and *[Eubacterium]_xylanophilum_group*.

## 4. Conclusions

FD is one of the most epidemic chronic functional gastrointestinal disorders. The quality of life for FD patients is affected by its reaction with the upper digestive tract and intestinal tract. Exploring safe natural substances that can attenuate the symptoms of FD has a significant meaning. In this study, the water extract of *Phyllanthus emblica* L. was verified to be capable of improving gastrointestinal motility alongside improving GE and SIT rate, thus improving the state of FD mice. Serum gastrointestinal hormones including GAS, MTL, VIP, and CCK were detected by ELISA kits, and the result showed that the serum GAS and serum MTL standards in APE groups were up-regulated and that the serum VIP was down-regulated compared to the model group. Gastrodynamic disorders are often characterized by variations in gastrointestinal hormone levels, and polyphenol substances and saccharides existing in APE may regulate the hormone secretion, therefore improving gastrointestinal motility. On the other hand, the gut microbiome structure and richness were also analyzed by 16s RNA sequencing to reveal the FD ameliorating effects of APE. The content of two primary phyla, Firmicutes and Bacteroidetes, were reversed in the model group in comparison to the normal group, and APE treatment restored this change. At the genus level, probiotics *Lactobacillus* and *Faecalibaculum* were elevated and harmful bacterial *Muribaculaceae*, *Alistipes*, and *[Eubacterium]_xylanophilum_group* were decreased in both the APEL and APEH groups.

## Figures and Tables

**Figure 1 foods-11-01491-f001:**
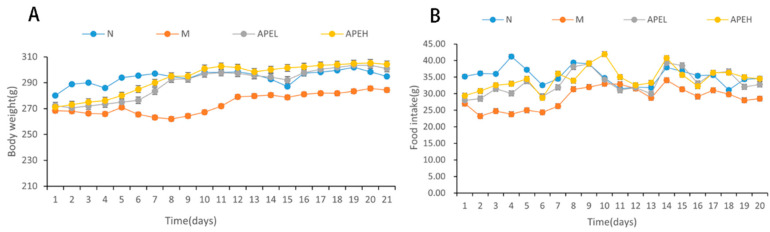
The body weight (**A**) and food intake (**B**) changes of mice in the process of APE administration. (N, normal group; M, model group; APEL, 150 mg/kg APE group; APEH, 300 mg/kg APE group).

**Figure 2 foods-11-01491-f002:**
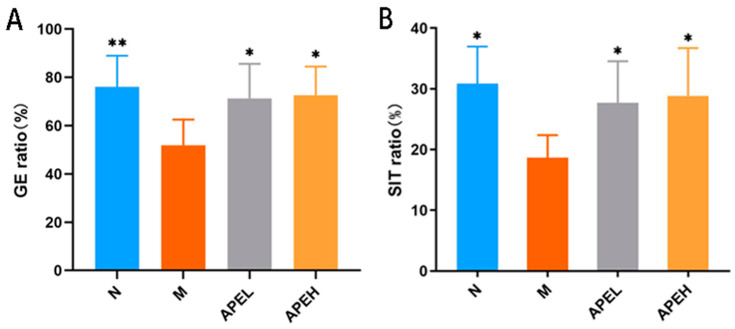
Effect of APE on GE ratio (**A**) and SIT ratio (**B**) in FD mice ((*) *p* < 0.05 and (**) *p* < 0.01 compared to the model group). (N, normal group; M, model group; APEL, 150 mg/kg APE group; APEH, 300 mg/kg APE group).

**Figure 3 foods-11-01491-f003:**
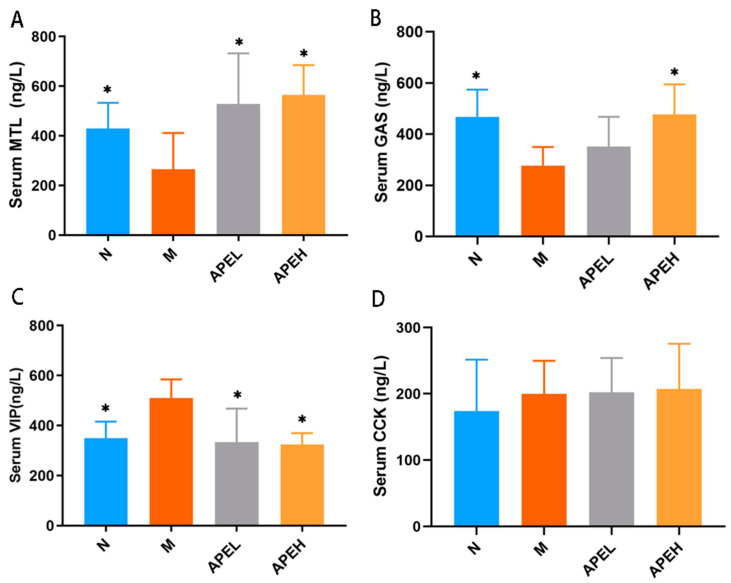
Effect of APE on serum gastrointestinal hormones MTL (**A**), GAS (**B**), VIP (**C**), and CCK (**D**) in FD mice ((*) *p* < 0.05 compared to the model group). (N, normal group; M, model group; APEL, 150 mg/kg APE group; APEH, 300 mg/kg APE group).

**Figure 4 foods-11-01491-f004:**
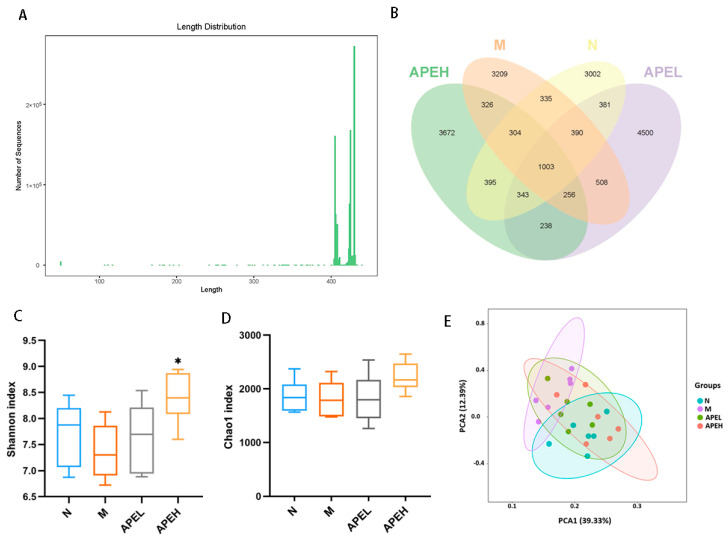
Effective sequence length distribution of intestinal microbial sequence tags in mice (**A**); Venn diagram of OTU distribution in the normal, model, APEL, and APEH groups (**B**); α diversity indicated by the (**C**) Shannon index and (**D**) Simpson index; PCA plots were used to visualize differences in weighted UniFrac distances of samples of OTUs from different groups (**E**). ((*) *p* < 0.05 compared to the model group). (N, normal group; M, model group; APEL, 150 mg/kg APE group; APEH, 300 mg/kg APE group).

**Figure 5 foods-11-01491-f005:**
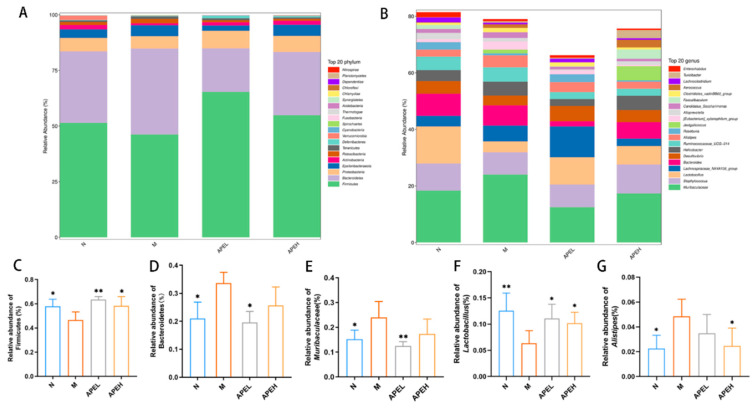
Relative abundance plots displaying the differences in the microbial community structure at the phylum level (top 20) (**A**) and genus level (top 20) (**B**); relative abundance of Firmicutes (**C**), Bacteroidetes (**D**), Muribaculaceae (**E**), Lactobacillus (**F**), and Alistipes (**G**). ((*) *p* < 0.05 and (**) *p* < 0.01 compared to the model group). (N, normal group; M, model group; APEL, 150 mg/kg APE group; APEH, 300 mg/kg APE group).

**Figure 6 foods-11-01491-f006:**
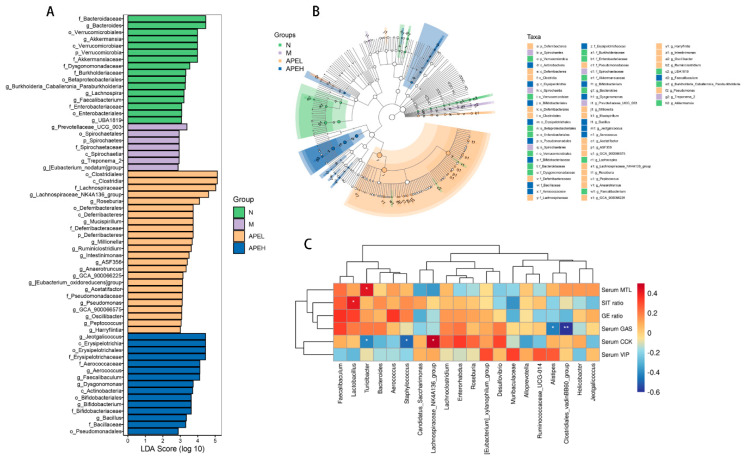
Microbial taxa discrepancies among normal, model, and APE groups (LDA scores of >2.0 and adjusted *p* values of <0.05). (**A**) Histogram and (**B**) cladogram. (**C**) Correlation analysis (*p* value of <0.05 and |R2| of >0.7) based on Spearman’s rank-order correlation between the top 20 abundant gut microorganisms at the genera level and biochemical criterion.

## Data Availability

The data presented in this study are available on request from the corresponding author. The data are not publicly available due to the funding requirement.

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
