# Peer review of "Aqueous Extract of Phyllanthus emblica L. Alleviates Functional Dyspepsia through Regulating Gastrointestinal Hormones and Gut Microbiome In Vivo"

_foods, 2022, doi:10.3390/foods11101491_

Round 1

Reviewer 1 Report

The manuscript "Aqueous extract of Phyllanthus emblica L. alleviates functional dyspepsia through regulating gastrointestinal hormone and gut microbiome in vivo" is an interesting article. It is well known that Phyllanthus emblica ameliorates functional dyspepsia (FD) and helpful in gastrointestinal diseases and in many more diseases. But as the authors explored the mechanism related to its beneficial effect in FD. Although, the manuscript is written well, some minor issues require authors' attention.

Line 42: "Helicobacter pylori" should be in italics.

Line 52-54: Correct and simplify the sentence.

Section 2.1: Please specify the amount of fresh fruit used for extraction and how much extract was obtained after lyophilization.

Figure 5: Phylum and Genus name is not clear.

Figure 6: Text in the figure should be readable.

Reviewer 2 Report

In the abstract: mention the experimental animals, the dose of APE and the duration of the intervention.

Scientific names should be italicized, revise the manuscript accordingly.

Kindly follow the Symbols and units format recommend on the journal website.

Line 124. Amplification should be amplicon

The composition and basic bioactivity of APE are required, even in terms of total phenolic content, and antioxidant capacity.

Follow the journal format to represent the figures and references.

Line 284, 285. In this study, the water extract of Phyllanthus emblica L. was verified possessing the activities of improving gastrointestinal motility and pepsin activity- Explain the possible mechanism.

Kindly correlate the microbial changes and disease reversing and discuss them in detail.

Reviewer 3 Report

The paper submitted by Lia and coworkers is focused on the evaluation of the influence of Phyllanthus emblica fruit extract on dyspepsia and the biodiversity of animal microbiota after oral ingestion. The paper is well prepared and most of the experiments are correctly planned and performed. However, some substantial corrections are needed in order to increase the scientific soundness of the paper.

1) more detailed info on the origin and authentication of plant material used for the study must be provided; how the material was identified? by whom? according to which literature? where the voucher specimen is deposited? what is the exact location of the collection site (GPS cordinates)?

2) the plant material used for any bioassay should be correctly characterized by the chemical analysis of the extract used for the bioassay. the authors should provide data from the phytochemical analysis of the extract. at least a simple chromatographic method like HPLC-UV or GC FID should be used for the characterization of the material. this is also important for the authentication of the material used for the preparation of the extract.

3) references section should be formatted according to journal's requirements
